# Interrater Reliability of ^99m^Tc-DMSA Scintigraphy Performed as Planar Scan vs. SPECT/Low Dose CT for Diagnosing Renal Scarring in Children

**DOI:** 10.3390/diagnostics10121101

**Published:** 2020-12-17

**Authors:** Hrefna Sæunn Einarsdóttir, Ronan Martin Griffin Berg, Lise Borgwardt

**Affiliations:** 1Department of Clinical Physiology, Nuclear Medicine & PET, Rigshospitalet, University of Copenhagen, Blegdamsvej 9, 2100 Copenhagen, Denmark; hrefnase@hotmail.com (H.S.E.); ronan.martin.griffin.berg@regionh.dk (R.M.G.B.); 2Department of Biomedical Sciences, Faculty of Health and Medical Sciences, University of Copenhagen, 2100 Copenhagen, Denmark; 3Centre for Physical Activity Research, University Hospital Rigshospitalet, 2100 Copenhagen, Denmark

**Keywords:** kidney disease, kidney function, paediatric, pyelonephritis, renal scarring

## Abstract

^99m^Tc-dimercaptosuccinic acid (DMSA) scintigraphy is currently the method of choice for assessing renal scarring in children, but it is not established whether conducting the scan as a single photon emission tomography combined with low-dose CT (SPECT/ldCT) scan provides additional diagnostic benefits when compared to conventional planar scintigraphy. In the present study, we evaluated the interrater reliability of DMSA SPECT/ldCT vs. planar DMSA scintigraphy for diagnosing renal scarring. Methods: Two nuclear medicine physicians blinded to patient data retrospectively analysed all paediatric ^99m^Tc-DMSA scintigraphes that were conducted in our department for the assessment of post pyelonephritis renal scarring between 2011 and 2016. All scintigraphies included both a planar scan and SPECT/ldCT, and were performed on either a Phillips Precedence 16 slice CT or a Siemens Symbia 16 slice CT. The readers were blinded to each other’s readings and to patient data, and assessed all scans dichotomously for evidence of renal scarring. For each scan, the readers further noted if they were confident in their interpretation. Results: A total of 46 pairs of planar SPECT/ldCT DMSA scans were included. The readers were unconfident about their interpretation of 40% of the planar scans and 5% of the SPECT/ldCT scans. The interrater agreement rate was 72% for planar scans and 91% for SPECT/ldCT, and the corresponding Cohen’s kappa values were 0.38 and 0.79. Conclusion: DMSA SPECT/ldCT is associated with higher reader confidence and interrater reliability than conventional planar DMSA scintigraphy for the assessment of post pyelonephritis renal scarring in children.

## 1. Introduction

Although ^99m^Tc-dimercaptosuccinic acid (DMSA) scintigraphy is widely considered the method of choice for detecting renal cortical scarring in children, notably after pyelonephritis [1,2], currently, conventional planar scintigraphy rather than single photon emission tomography combined with low-dose CT (SPECT/ldCT) is recommended [3]. Diagnostically, SPECT/ldCT nonetheless has several potential benefits in that it provides three-dimensional images of the renal parenchyma in toto, while only approximately three-quarters of the kidney parenchyma are visualized appropriately by planar scintigraphy. Some studies have thus reported that SPECT/ldCT increases sensitivity for detecting renal cortical defects [4,5], while others found no apparent advantage over planar scintigraphy [6], and the exact additional diagnostic benefit of SPECT/ldCT thus remains to be established.

In the present study, we assessed the interrater reliability of DMSA scintigraphy for detecting renal scarring in children when conducted as a planar scan vs. SPECT/ldCT and hypothesised that the latter would yield a higher interrater reliability.

## 2. Materials and Methods

We retrospectively identified all paediatric patients (age 0–18 years) that were referred to the Department of Clinical Physiology, Nuclear Medicine and PET, Rigshospitalet between 2011 and 2016 for the assessment of renal scarring 4–6 months post pyelonephritis. According to our local protocol, all patients concurrently underwent planar DMSA scintigraphy and SPECT/ldCT, and were injected with up to 60 MBq ^99m^Tc-DMSA according to the EANM dosage calculator [7]. All procedures performed were in concordance with the ethical standards of the institutional and national research committee (J.nr: 2007-58-0006. AHH-2016-094, I-Suite nr.:05158) and with the 1964 Helsinki declaration and its later amendments or comparable ethical standard. Informed consent was obtained from all individual participants included in the study. Findings relating to split renal function assessments from most of these patients have been published elsewhere [8].

### 2.1. Scan Protocols and Reconstruction Methods

All examinations were performed on either a Precedence 16 slice CT (Philips Healthcare, Best, The Netherlands) or a Symbia 16 slice CT (Siemens, Erlangen, Germany). In the following, Philips and Siemens parameters are separated by a slash. Heads were mounted with Low Energy General/All Purpose collimators. The energy window was set at 140 keV with 20/15% width.

Planar scans were performed in the anterior (A) and posterior (P) position as close as possible to the patient with an acquisition time of 15 min. SPECT was with two heads, and over 180 degrees for each head and 120/128 angles in step and shoot mode. Total acquisition time was approximately 22 min with 20 s per angle. SPECT was followed by ldCT that was performed only on a belt over the kidneys with 80 kVp and 20 mAs, yielding an effective dose of approximately 0.2 mSv. The radiation dose per examination without low-dose CT of the kidneys, irrespective of age, is approximately 1 mSv, provided that the dose is correctly adapted to body size. The radiation dose per examination including SPECT with low-dose CT is then approximately 1.2 mSv. Iterative reconstruction was performed on the scanners including CT-based attenuation correction including resolution modelling (Astonish/Flash3D).

### 2.2. DMSA Scintigraphy Readings

Two board-certified nuclear medicine specialists, one of which was a dedicated paediatric nuclear medicine physician (Reader 1) and the other a general nuclear medicine physician (Reader 2) read all DMSA scintigraphies. The readers were blinded to patient data, outcomes, and each other’s readings, and each reader assessed planar and SPECT/ldCT scans from the same patients on separate occasions. Distinct cortical hypoactive areas were considered evidence of renal scarring [2,3]. For each scan, the readers further noted if they felt unconfident about their interpretation.

### 2.3. Statistics

Continuous data are presented as median (IQR), while categorical data are presented in %. All scans were classified dichotomously by each rater as showing evidence or no evidence of renal scarring. Cohen’s Kappa coefficient (with 95% CI) was calculated, considering values of 0 as no agreement, above 0 but ≤20 as slight, 0.21–0.41 as fair, 0.41–0.60 as moderate, 0.61–0.80 as substantial, and 0.81 to 1.00 excellent agreement [9]. All analyses were performed using SAS statistical software version 9.2 (SAS Institute Inc., Cary, NC, USA).

## 3. Results

We identified a total of 46 DMSA scintigraphies performed in 46 children that all included both planar scans and SPECT/ldCT. The patients’ median age was 6 years and two months (IQR: 1 year and 8 months to 10 years and 4 months), and the male/female ratio was 11/89. Representative DMSA scans are provided in Figure 1 and Figure 2.

The results of the readers’ assessments are summarized in Table 1. Together, the readers were unconfident about their interpretation of 40% of the planar scans and 5% of the SPECT/ldCT scans. When a reader interpreted a planar scan as showing no evidence of renal scarring, but was unconfident about the diagnosis, the corresponding SPECT/ldCT was interpreted as showing evidence of renal scarring in 37% of the cases. In contrast, when a reader interpreted a planar scan as showing evidence of renal scarring, but was unconfident about the diagnosis, the corresponding SPECT/ldCT was interpreted as showing no evidence of renal scarring in 35% of the cases.

The agreement rate between readers for interpreting planar scans was 72% while it was 91% for SPECT/ldCT, and the corresponding Cohen’s kappa values were 0.38 (95% CI: 0.15–0.60) and 0.79 (95% CI: 0.59–0.99).

## 4. Discussion

The main finding of the present study is that the interrater reliability DMSA scintigraphy for diagnosing post pyelonephritis renal scarring in children is much higher at a kappa value of 0.79, corresponding to substantial agreement, when conducted as a SPECT/ldCT rather than a conventional planar scan which yielded a kappa value of 0.38, corresponding to only moderate agreement. Furthermore, our findings suggest that readers may be more confident when interpreting SPECT/ldCT than planar scans.

The findings that readers were less confident in their diagnosis and disagreed more when assessing planar scintigraphy is in accordance with a previous study on children where planar DMSA scintigraphy was compared to SPECT, and this tended to be more pronounced for inexperienced readers [10]. In contrast to our findings, another study reported similar overall kappa values between planar scintigraphy and SPECT among three readers, i.e., 0.59 and 0.57, respectively, corresponding to moderate agreement in both cases [11]. However, these findings are difficult to interpret, because the kappa values between pairs of readers ranged from negative to 1.00 [11]. Furthermore, neither of the abovementioned studies supplemented SPECT with a ldCT, which may add further to reader confidence by delineating renal structures, while also increasing specificity by providing diagnostic clues to lesions that do not represent scars [3], such as renal abscess, cysts, duplex kidney, hydronephrosis, and persistent foetal lobulation (Figure 2).

It is also important to mention that oblique static images are recommended as a supplement to the planar images and should be part of the evaluation of the diagnostic value. However, they are very often not possible to perform because the child reacts to being positioned in the oblique position; it is thus not an option in most daily routine, and therefore, not performed in this study.

We believe that our findings support the use of DMSA SPECT/ldCT for assessing renal scarring in children, and that this should be considered in future guidelines. Among the major concerns for implementing this in standard practice is, however, both the potentially increased need for sedation and the slightly increased higher radiation dose due to the ldCT [3]. Although we did not systematically record the need for additional sedation due to the SPECT/ldCT, this was only necessary in a few patients in the age group of 1–3 years. When considering the potential clinical consequences of wrongly classifying a child as showing no evidence of renal scarring after an episode of pyelonephritis [1,2], the additional radiation dose of 0.2 mSv due to the ldCT is, in our opinion, acceptable in this context.

In conclusion, we found that the interrater reliability of SPECT/ldCT DMSA scintigraphy is substantial and is superior to that of conventional scintigraphy for detecting renal scarring in children.

## Figures and Tables

**Figure 1 diagnostics-10-01101-f001:**
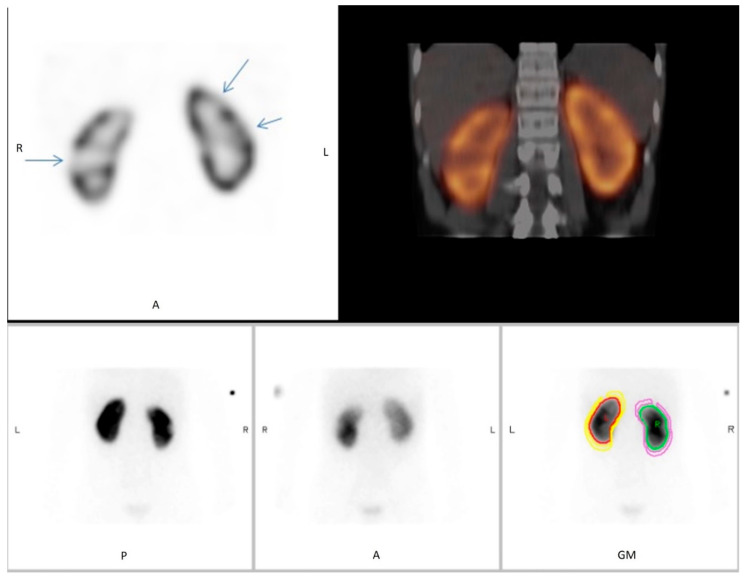
^99m^Tc-dimercaptosuccinic acid ^99m^Tc-DMSA scintigraphy as planar scan and SPECT/low-dose CT (SPECT/ldCT) in a 13-year-old boy. Evaluation scan conducted 3 months after pyelonephritis. On the planar scintigraphy, two defects that are suspicious of infarction are evident on the lateral and upper pole of the right kidney. However, when supplementing with SPECT/low-dose CT, additional infarction is unveiled in the left kidney not seen on planar scintigraphy. Upper panel: SPECT (**left**), SPECT/ldCT (**right**). Lower panel: planar scan, posterior view (**left**), anterior view (**middle**), geometric view (**right**).

**Figure 2 diagnostics-10-01101-f002:**
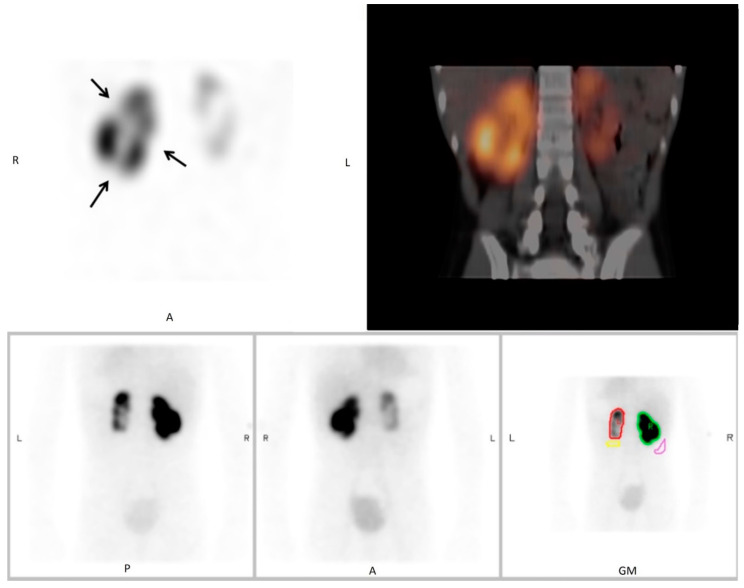
^99m^Tc-DMSA scintigraphy as planar scan and SPECT/low-dose CT (SPECT/ldCT) in a 10-year-old girl. Evaluation scan conducted 3 months after pyelonephritis. Planar scintigraphy shows an atrophic left kidney and a normal right kidney with a small defect that was interpreted as persistent foetal lobulation due to incomplete fusion of the developing renal lobules. However, SPECT/low-dose CT shows that the defect is due to scarring, since cortical atrophy and irregular scarring are seen on CT. Upper panel: SPECT (**left**), SPECT/ldCT (**right**). Lower panel: planar scan, posterior view (**left**), anterior view (**middle**), geometric view (**right**).

**Table 1 diagnostics-10-01101-t001:** Assessments of DMSA scintigraphies by Reader 1 and Reader 2.

	Reader 1	Reader 2
**Planar Scintigraphy**		
Evidence of renal scarring	20 (43%)	7 (15%)
No evidence of renal scarring	26 (57%)	39 (85%)
Unconfident about diagnosis	16 (35%)	20 (43%)
**SPECT/ldCT**		
Evidence of renal scarring	13 (28%)	12 (26%)
No evidence of renal scarring	33 (72%)	34 (74%)
Unconfident about diagnosis	1 (2%)	4 (9%)
**Planar Scintigraphy vs. SPECT/ldCT**		
Concordant diagnoses between modalities	37 (85%)	35 (76%)
Evidence of renal scarring	12 (32%)	4 (11%)
No evidence of renal scarring	25 (68%)	31 (89%)
Discordant diagnoses between modalities	9 (15%)	11 (24%)
Evidence of renal scarring on planar scintigraphy No evidence on SPECT/ldCT	8 (89%)	3 (27%)
No evidence of renal scarring on planar scintigraphy Evidence on SPECT/ldCT	1 (11%)	8 (73%)

The underline: separate concordant from discordant.

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
