# Peer review of "Interrater Reliability of 99mTc-DMSA Scintigraphy Performed as Planar Scan vs. SPECT/Low Dose CT for Diagnosing Renal Scarring in Children"

_diagnostics, 2020, doi:10.3390/diagnostics10121101_

Round 1

Reviewer 1 Report

The article is well structured and the results support the conclusions.

Some minor typos to correct.

The article tackles a question that is relevant to the daily practice of Paediatric Nuclear Medicine. DMSA scan is one of the most performed modalities in this setting, and there has been a long-standing debate on the pros and cons of tomographic imaging vs. standard planar imaging. The data presented in the article are well-described. They correctly define the possible role of DMSA SPET-CT in increasing the clinical confidence in diagnosing post-pyelonephritic scars, particularly when the reader is not fully dedicated to paediatric nuclear medicine. The Authors address the possible increased need for sedation, and they offer their experience as a valuable starting point. In summary, the article will be of interest to most NM specialists involved in Paediatric Nuclear Medicine and could be the base for other studies in the future.

Author Response

The article is well structured and the results support the conclusions. In summary, the article will be of interest to most NM specialists involved in Paediatric Nuclear Medicine and could be the base for other studies in the future.

Author response: Thank you very much!

  1. Comment from Reviewer 1 suggesting to correct some minor typos

Author response: Thank you for pointing this out. The reviewer is correct, and we have corrected the typos.

The revised text reads as follows “Although 99mTc-dimercaptosuccinic acid (DMSA) scintigraphy is widely considered the method of choice for detecting renal cortical scarring in children, notably after pyelonephritis [1,2], and currently conventional planar scintigraphy rather than single photon emission tomography combined with low dose-CT (SPECT/ldCT) is recommended [3]. Diagnostically, SPECT/ldCT nonetheless has several potential benefits in that it provides 3-dimensional images of the renal parenchyma in toto, while only approximately three quarters of the kidney parenchyma are visualized appropriately by planar scintigraphy.”

Reviewer 2 Report

Thank you very much for this interesting manuscript. The results are exciting and may lead to a revision of the current EANM guidelines concerning the DMSA scintigraphy on children.

I am missing the total amount of radiation dose that the two examined techniques exhibit: planar scintigraphy versus SPECT/ldCT DMSA with a subsequent comment on the radiation dose difference between the 2 techniques. 

Otherwise, the manuscript is clear and to the point. Well done!

Author Response

The results are exciting and may lead to a revision of the current EANM guidelines concerning the DMSA scintigraphy on children

Author response: Thank you very much!

  1. Comment from Reviewer 2 suggesting to add the total amount of radiation dose that the two examined techniques exhibit: planar scintigraphy versus SPECT/ldCT DMSA with a subsequent comment on the radiation dose difference between the 2 techniques. 

Author response: Thank you for pointing this out. The reviewer is correct, and we have incorporated the suggestions

The revised text reads as follows: ”SPECT was followed by ldCT that was performed only in a belt over the kidneys with 80 kVp and 20 mAs yielding an effective dose of approximately 0.2 mSv. The radiation dose per examination without lowdose CT of the kidneys, irrespective of age, is approximately 1 mSv, provided that the dose is correctly adapted to body size. The radiation dose per examination including SPECT with lowdose CT is then approximately 1.2 mSv